# Local Delivery of Immunomodulatory Antibodies for Gastrointestinal Tumors

**DOI:** 10.3390/cancers15082352

**Published:** 2023-04-18

**Authors:** Noelia Silva-Pilipich, Ángela Covo-Vergara, Cristian Smerdou

**Affiliations:** 1Division of Gene Therapy and Regulation of Gene Expression, Cima Universidad de Navarra, 31008 Pamplona, Spain; acovo@alumni.unav.es; 2Instituto de Investigación Sanitaria de Navarra (IdISNA), Cancer Center Clínica Universidad de Navarra (CCUN), 31008 Pamplona, Spain

**Keywords:** gastrointestinal tumors, immunotherapy, immune checkpoint blockade, viral vectors, non-viral vectors

## Abstract

**Simple Summary:**

Many types of gastrointestinal tumors, such as gastric, colorectal, and pancreatic cancer, do not respond well to immunotherapies based on the use of antibodies against immune checkpoints, which are injected systemically into patients and generate frequent adverse effects. This review focuses on alternative ways to deliver immunostimulatory antibodies based on gene therapy vectors able to produce them locally at the tumor site. In particular, the use of modified viruses as vectors can induce local inflammation, which contributes to generating stronger antitumor responses. Many preclinical studies show that gastrointestinal tumors could respond better to immunotherapy by using these novel delivery approaches.

**Abstract:**

Cancer therapy has experienced a breakthrough with the use of immune checkpoint inhibitors (ICIs) based on monoclonal antibodies (mAbs), which are able to unleash immune responses against tumors refractory to other therapies. Despite the great advancement that ICIs represent, most patients with gastrointestinal tumors have not benefited from this therapy. In addition, ICIs often induce adverse effects that are related to their systemic use. Local administration of ICIs in tumors could concentrate their effect in the malignant tissue and provide a higher safety profile. A new and attractive approach for local delivery of ICIs is the use of gene therapy vectors to express these blocking antibodies in tumor cells. Several vectors have been evaluated in preclinical models of gastrointestinal tumors to express ICIs against PD-1, PD-L1, and CTLA-4, among other immune checkpoints, with promising results. Vectors used in these settings include oncolytic viruses, self-replicating RNA vectors, and non-replicative viral and non-viral vectors. The use of viral vectors, especially when they have replication capacity, provides an additional adjuvant effect that has been shown to enhance antitumor responses. This review covers the most recent studies involving the use of gene therapy vectors to deliver ICIs to gastrointestinal tumors.

## 1. Introduction

Tumors in the gastrointestinal (GI) tract include esophageal cancer, gastric cancer, colorectal cancer (CRC), liver cancer, and pancreatic cancer. As a group, they represent a major public health burden and accounted for 26% of the global cancer incidence and 35% of cancer-related deaths in 2018 [1]. CRC is the most frequent GI cancer, ranking third in terms of incidence and second in mortality of all cancer types [2]. The GI cancer burden is greater in countries with high human development indices, and risk factors include genetic and environmental factors such as unhealthy diets, sedentary lifestyles, alcohol consumption, and smoking [3]. The global incidence and mortality related to GI cancers are predicted to increase by 58% and 66%, respectively, by 2040 [4]. Despite efforts in disease prevention and improvements in screening methods for early diagnosis, especially for CRC [5], this group of malignancies will continue to pose a major challenge in the foreseeable future.

One of the main challenges of GI cancers is that early diagnosis is minimal since symptoms are non-specific. Conventional therapeutic options include surgical resection, chemotherapy, and radiotherapy; however, patients with advanced disease have a poor prognosis [1]. Immunotherapies have become promising new strategies for the treatment of some cancers, giving hope to patients with unresectable or advanced diseases who otherwise would have few options. Immunotherapies comprise an array of approaches aiming to exploit the patient’s immune system to specifically eliminate tumor cells, and include the use of immunomodulatory antibodies, cancer vaccines, cytokines, adoptive cell therapy, and engineered T cells [6]. Immune checkpoint blockade (ICB) constitutes one of the most widely employed types of immunotherapies and has shown remarkable therapeutic benefit in various types of solid tumors [7]. However, clinical benefit in patients with GI cancers is mainly limited to microsatellite instability-high (MSI-H) GI cancers, while the rest of the tumors remain largely unresponsive to immunotherapies. Several combination strategies to improve the efficacy of ICB in GI cancers are under clinical evaluation, and results are eagerly awaited. Recent comprehensive reviews have addressed the latest advances in ICB, alone or in combination with other therapeutic strategies, for the treatment of GI tumors [6,8,9].

Another important point to be considered is the urge to increase the safety of treatments based on immunomodulatory antibodies. In this regard, optimization of administration schemes for immune checkpoint inhibitors (ICIs) that reduce the doses that patients receive could have a positive impact on the safety of these therapies. Exploiting gene therapy approaches to deliver therapeutic molecules to the tumor area is an interesting alternative to conventional systemic administration since it could promote their accumulation in the tumor microenvironment (TME), potentially increasing the therapeutic effect and minimizing damage to healthy tissues. In this review, we describe the latest developments in the field of gene therapy to deliver ICIs based on antibodies to GI tumors and comment on combination strategies that are being evaluated with ICB therapies.

## 2. Immune Checkpoint Blockade in Gastrointestinal Cancers

Immune checkpoints are receptors with inhibitory functions that are expressed in different immune cells and contribute to regulating immune responses [10]. It has been described that sustained exposure of the T cells to their antigen leads to a progressive loss of their effector functions and proliferative potential, resulting in an “exhausted” phenotype characterized by the upregulation of multiple immune checkpoints [11]. In the context of cancer, the expression of immune checkpoints by T cells and other immune cells and the cognate ligands by tumor cells and other cells in the TME promote the inhibition of the antitumor immune response and constitute one mechanism of tumor immune escape. Immune checkpoints can be targeted for therapeutic purposes using ICIs, with the aim of reinvigorating exhausted T cells and unleashing dormant antitumor immune responses [12,13].

Several immune checkpoints have been described and constitute promising therapeutic targets for cancer immunotherapy, including CTLA-4 (cytotoxic T lymphocyte antigen-4) [14], PD-1 (programmed death-1) [15], LAG-3 (lymphocyte activation gene-3) [16], TIM-3 (T-cell immunoglobulin and mucin domain-3) [17], BTLA (B and T lymphocyte attenuator) [18], TIGIT (T-cell immunoreceptor with immunoglobulin and ITIM domain) [19], and VISTA (V-domain Ig suppressor of T-cell activation) [20]. The most clinically advanced inhibitors are those directed against CTLA-4 and the PD-1/PD-L1 axis, with several monoclonal antibodies (mAbs) already on the market for the treatment of different solid tumors [7].

CTLA-4 and PD-1 pathways differ in timing and the main anatomic locations in which they exert their functions. CTLA-4 is constitutively expressed at high levels by regulatory T cells (Tregs), and it is upregulated soon after activation in effector T cells. By binding to CD80 and CD86, it prevents the hyperactivation of T cells, mainly during the priming phase in the lymph nodes [14,21]. Activated T cells also upregulate PD-1, but at a later time point, and upon binding to its ligands, PD-1 inhibits the positive signaling of the T-cell receptor (TCR) and leads to a decrease in cytokine production, proliferation, and survival of T cells. Two ligands have been described for PD-1, programmed death ligand 1 (PD-L1) and 2 (PD-L2). PD-L1 is upregulated in a variety of cells in response to inflammation, especially by interferon-gamma (IFN-γ) signaling; hence, this pathway works as a negative feedback mechanism preventing tissue damage from hyperactivation or autoreactivity of T cells [22]. Targeting the PD-1/PD-L1 pathway has shown a more favorable response/toxicity ratio compared to the blockade of CTLA-4, which has led to the approval of various antibodies against PD-1 and PD-L1 for the treatment of different malignancies, including CRC, gastric cancer, esophageal cancer, and hepatocellular carcinoma (HCC) [7,23]. Although many patients benefit from ICB and some achieve long-term tumor regressions, the majority of patients fail to respond or respond only transiently. In addition, these treatments frequently induce immune-related adverse effects (irAEs) that can result in long-term complications and could potentially affect any organ, including the dermatologic, gastrointestinal, pulmonary, endocrine, and cardiovascular systems [24].

Understanding primary and secondary mechanisms of resistance to ICB in the context of GI tumors is key to optimizing the therapeutic strategies and achieving maximal clinical benefit. With our increasing understanding of tumor biology and immunology, different factors have been pointed to as potential contributors to the failure of ICB, including (i) gut microbiota composition, (ii) poor penetration and heterogeneous distribution of therapeutic mAbs in the tumor mass, (iii) strong immunosuppression in the TME, and (iv) low tumor immunogenicity or impaired antigen presentation. Different studies have attempted to tackle these limitations by designing combination strategies aimed at modulating the gut microbiota, concentrating the immunomodulatory mAbs in the TME by modifying the route of administration or introducing tumor-targeting moieties, enhancing inflammation in the TME, and boosting tumor-antigen release and presentation. Promising results from preclinical studies have led to the clinical evaluation of some of these strategies. In particular, the use of gene therapy vectors opens a new avenue in the field of ICI delivery and immunomodulation.

## 3. Microbiota Impacts on Immunotherapy Efficacy

The gut microbiota is the diverse and constantly evolving collection of microorganisms present in the gastrointestinal tract, including bacteria, archaea, viruses, and unicellular eukaryotes [25]. Accumulating evidence supports the important role of gut microbiota composition in cancer development and response to therapies, including ICB, with a pivotal role in GI cancers [26,27]. The exact biological mechanisms explaining this relationship are not fully understood; however, some hypotheses have been proposed: (i) the presence of certain bacterial metabolites that can have tumor promoting or antitumor effects [28], (ii) molecular mimicry between tumor-associated antigens and microbiota-derived epitopes [29,30,31], and (iii) bacterial translocation into the tumor site, where they could modulate immune responses [32].

Different tumors are known to be colonized by bacteria [33,34,35,36,37]. A recent study has shown that the tumor microbiome varies depending on the tumor type and that bacteria can be found intracellularly in both tumor and immune cells [38]. In addition, bacteria-derived antigens can be presented in the HLA molecules of tumor cells and induce immune activation, which could represent one of the mechanisms behind their influence on treatment outcomes [39]. In the case of CRC, the influence of microbiota on cancer development and progression is of particular interest since the colon contains trillions of bacteria, and patients with sporadic CRC frequently show abnormalities in their gut microbiomes [40]. Beyond CRC, gut and tumor microbiota composition can also influence the development of different malignancies such as HCC, pancreatic, breast, and lung cancer [41,42,43,44].

One known example of the relationship between cancer and microbiota is represented by *Helicobacter pylori* (*H. pylori*). For a long time, a link between *H. pylori* infection and gastric cancer development has been recognized [45]. This bacterium has been described to modify the microenvironment within the gastric mucosa, affecting different kinds of tumor stromal cells and favoring immune escape and cancer progression. Furthermore, *H. pylori* increases PD-L1 expression in gastric cancer, leading to resistance to immunotherapies [46]. *H. pylori* has also been associated with a higher risk of pancreatic cancer [47], which is characterized by harboring pathogenic bacteria that induce immunosuppression in the TME. Ablation of the microbiome in orthotopic models of pancreatic cancer relieved immunosuppression and enhanced T-cell activation. Notably, the combination of microbial ablation with ICB synergistically decreased tumor size and further enhanced the activation of T cells [48].

Bacteria can also aid in the antitumor immune responses, which has been demonstrated by the fact that the use of broad-spectrum antibiotics prior to ICB treatment decreases the therapeutic efficacy in mouse models [49,50,51] and has a detrimental effect on the overall survival and objective response rates in patients with different cancer types as observed in retrospective analysis [52,53,54]. The use of broad-spectrum antibiotics prior to ICB therapy is currently discouraged [55], although it would be possible to use targeted antibiotics to reduce certain species of bacteria that promote immunosuppression [56,57].

A plethora of recent studies have addressed how microbiota composition can affect the outcome of ICB immunotherapy. The first evidence of this association was noticed in melanoma mouse models, where it was found that the gut microbiota modulates the response to anti-CTLA-4 [49] and anti-PD-L1 [58] antibody therapies. These studies identified distinct *Bacteroides* and *Bifidobacterium* species associated with the enhancement of ICB efficacy. Similar associations have been reported in GI cancers. For example, a study in patients with unresectable HCC and advanced biliary tract cancers who progressed after first-line chemotherapy showed an association between the gut microbiome and responses to anti-PD-1 therapy, with enrichment in taxa associated with energy metabolism in responder patients [59]. Another cohort study in patients with HCC revealed a pattern not only in the microbiota composition but also in microbial metabolites. In particular, secondary bile acids were significantly higher in patients who achieved objective responses [60]. Some studies have also demonstrated the protective effect of *Lactobacillus* and *Enterococcus faecium* against CRC, thanks to the secretion of exopolysaccharides and orthologs of peptidoglycan hydrolase secreted antigen A, respectively, which have immunostimulatory activity and enhance ICB efficacy [61,62]. Other bacterial metabolites, such as trimethylamine N-oxide (TMAO), potentiate the type I interferon (IFN-I) pathway and lead to a significant tumor reduction in pancreatic ductal adenocarcinoma mouse models when combined with ICB [63].

Despite the compelling evidence on the association between the microbiome and the outcome of immunotherapy, especially that based on ICB, further studies are needed to achieve a better understanding of this intricate crosstalk. The identification of bacteria associated with cancer development, progression, and response to therapies is crucial for the design of precision medicine based on microbiota modulation. Several strategies could be used to increase immunotherapy efficacy, including fecal microbiota transplantation (FMT), dietary intervention, targeted antibiotics, probiotics, bacteriophage therapy, and bacterial metabolite supplementation [55,64,65]. FMT has demonstrated to be a feasible strategy to treat recurrent *Clostridium difficile* infection, and it is being evaluated for the treatment of other diseases that are suspected to be originated by or influenced by gut dysbiosis [66]. FMT from healthy donors or ICB-responder patients could become a practical and safe strategy to increase ICB efficacy in resistant patients, and it has also shown to be useful for the management of refractory ICB-induced colitis [67,68]. However, standardization of protocols for donor selection, stool sample formulation, and preparation of the recipient patients is needed. In addition, concerns regarding microbiota composition variability among donors and the potential risk of disease transmission suggest that administration of specific probiotic formulations that have proven beneficial for ICB therapy could be a safer and more standardizable approach [26]. In terms of dietary intervention, strategies to increase microbiota diversity and frequency of beneficial strains involve consensus good nutritional practices, such as increasing intake of dietary fiber, whole grains, plant protein, and fermented foods, and decreasing red meat consumption and added sugars. For example, the combination of anti-PD-1 treatment with pectin, a widely consumed soluble fiber, restored anti-PD-1 efficacy in tumor-bearing mice humanized with gut microbiota from patients with CRC by beneficially regulating their composition and diversity. In particular, immunomodulatory butyrate-producing bacteria such as *Lactobacillaceae*, *Bifidobacteriaceae*, *Erysipelotrichaceae*, and *Ruminococcaceae* were enriched upon pectin and anti-PD1 combination, which promoted T-cell infiltration in the TME and enhanced anti-PD-1 efficacy [69]. Other less conventional strategies such as ketogenic diets, calorie restriction, and short-term starvation have shown to be beneficial in animal models, although more studies are needed in the clinical setting [55,70].

Finally, some studies have reported that patients who develop intestinal inflammation due to ICB are more likely to respond to this therapy [71,72,73]. It is reasonable to think that this damage facilitates bacterial translocation across the gut lumen and allows bacteria to colonize distant tissues, preferentially tumors [32]. This preference is likely a result of the unique properties of the TME, including reduced immunosurveillance and hypoxia [74]. If this model is correct, it may imply that modulation of the gut microbiota may be insufficient for patients who do not develop gastrointestinal irAEs. This could be solved by local injection of the beneficial bacteria strains in the tumor mass or systemic administration using extremely low doses to avoid sepsis. Furthermore, bacteria are versatile platforms that can be engineered to increase safety and therapeutic efficacy by different strategies, including increased tumor targeting, delivery of therapeutic payloads into the TME, and reprograming of the immune system [75]. For example, bacteria can be engineered to express ICIs based on antibody fragments such as Fab fragments, single-chain variable fragments (scFv), or single-domain antibodies derived from camelids (nanobodies), since these molecules are adequately expressed by prokaryotic cells [76]. In addition, programmable bacteria can be designed to selectively grow or release therapeutic payloads in response to tumor-associated stimuli [77,78], as has been demonstrated for the delivery of nanobodies against PD-L1 and CTLA-4 in the CT26 subcutaneous tumor model [77]. Despite the great potential of bacterial-based cancer therapy, relevant issues remain to be addressed regarding safety, including undesirable bacterial infection, septic shock, tumor-lysis syndrome, the acquisition of antibiotic-resistant genes, and the reversion of attenuation mutations [79].

## 4. Local Delivery of Immunomodulatory Antibodies

Solid tumors are complex tissues composed of multiple cell types and extracellular components. The TME plays decisive roles in tumor progression, metastasis, and response to therapies [80]. Tumors present an abnormal extracellular matrix (ECM), with an increased accumulation of hyaluronic acid, collagen, and fibronectin, as well as proteases that contribute to its remodeling and influence the angiogenic process [81]. As a result, tumors are characterized by a high interstitial pressure, aberrant vasculature, and a dynamic composition of the ECM that can influence not only tumor development but also the response to therapies since it constitutes a physical barrier that inhibits the infiltration of immune cells and drugs [82].

Inefficient and heterogenous distribution of therapeutic mAbs in the TME could contribute to the development of acquired resistance and treatment failure, since areas of the tumor that are more difficult to penetrate may receive only marginal doses of mAbs [83,84]. This problem is particularly evident due to the large size of mAbs and may be ameliorated by using smaller antibody fragments such as scFv or nanobodies, which have shown an improved rate of tumor uptake and a more homogenous intratumoral distribution [85,86,87]. Furthermore, preclinical observations indicate that a local route of delivery could be more efficient at initiating, enhancing, and maintaining a strong antitumor T-cell response [88]. Local delivery can potentially increase the accumulation of antibodies in the tumor bed, thus enhancing their therapeutic efficacy while decreasing systemic exposure that could give rise to irAEs. Different technologies for intratumoral delivery of mAbs have been evaluated, such as injectable hydrogels, implantable biomaterials, and microneedles [89,90]. For tumors that are not easily accessible, systemic administration of modified therapeutic mAbs that selectively accumulate in the TME, for example, by fusing them to ECM-anchoring domains [91] or designing antibody prodrugs (probodies) that are conditionally activated in tumors are strategies that could be used [92]. Other approaches include the use of actively targeted nanoparticles for tumors [93] or more biological systems, such as engineered erythrocytes. For example, modified erythrocytes loaded with super-paramagnetic nanoparticles and a mAb against FAT1, a CRC-associated protein, were able to accumulate magnetically in the tumor area, where the delivery of the therapeutic mAb to the cytoplasm of tumor cells was facilitated by the addition of a fusogenic protein on the surface of the erythrocytes [94]. This innovative strategy could potentially be used to inhibit intracellular proteins during cellular trafficking, for example in the case of PD-L1 recycling in tumor cells [95].

Despite the fact that these strategies show great potential, clinical-grade proteins are difficult and expensive to produce, and high doses of mAbs are required for each patient. In this regard, gene therapy would be an interesting alternative approach for inducing the expression of therapeutic molecules within the tumors. The wide array of vectors available allows for the design of different strategies that take advantage of their properties [96]. For example, long-term expression vectors could theoretically avoid the need for repeated dosing of antibodies, providing a sustained level of the therapeutic agent for the entire course of treatment. In addition, the biological effects of the vector itself can be exploited to induce changes in the TME, for example, by inducing inflammation and tumor cell death, which could sensitize tumors to immunomodulatory antibodies.

Viral vectors are the most commonly used delivery vehicles in pre-clinical and clinical settings due to their remarkable gene delivery efficiency [97], although their use raises some concerns regarding their safety, pre-existing antibodies, poor packaging capacity, and high production costs [98,99,100]. Non-viral vectors are also under active investigation since they present a more favorable safety profile, can deliver larger genetic payloads, and are typically easier and less expensive to produce than viral vectors. However, their gene transfer efficiency, specificity for the target cells, and duration of gene expression are usually lower than those of viral vectors [101]. Several types of viral and non-viral vectors able to express ICIs have been tested in different preclinical models of GI tumors with promising results (Figure 1).

In the clinical setting, after decades of research and numerous clinical trials, gene therapy is currently at an exciting point in which several therapies are receiving regulatory approvals for a variety of life-threatening diseases [102] and the delivery of therapeutic antibodies may be getting closer to clinical implementation.

## 5. Viral Vectors to Deliver Immunomodulatory Antibodies

Vectors used in gene therapy have traditionally been based on viruses, taking advantage of their remarkable ability to transfer their genome into the host cell. To date, viral vectors are still the first choice in clinical trials, with lentivirus, adenovirus, and adeno-associated viruses (AAVs) being the most widely used vectors (accounting for 50% of the total clinical trials involving gene therapy) [103]. In the context of cancer, viruses can be used as gene delivery vehicles alone or as oncolytic viruses (OVs). The first group encompasses viral vectors that have been engineered to prevent their replication and are used as vehicles to deliver therapeutic genes without inducing an active infection in the cells. Since they do not propagate, they tend to be less immunogenic and usually allow expression of the transgene for longer periods. On the contrary, OVs retain their replication ability, holding a closer resemblance to their natural counterparts. OVs preferentially replicate in tumor cells, which are often deficient in antiviral defense mechanisms due to deregulation of different pathways such as interferon, Wnt, and Ras. They induce not only the lysis of tumor cells but also the activation of the immune system due to the release of viral components, damage-associated molecular patterns (DAMPs), tumor-associated antigens (TAAs), and neoantigens in the TME. Therefore, they can be considered adjuvants for in situ cancer vaccination. On top of that, OVs can also be employed as gene delivery vehicles by arming them with therapeutic genes such as immunomodulatory antibodies or cytokines, further modulating the TME [104,105].

At the interface between non-propagating vectors and OVs, there is a third group of vectors derived from alphaviruses. These vectors, which are based on self-amplifying RNA, could theoretically be used as propagating viruses; however, unlike OVs, they present a wide tropism and no tumor-targeting preference, hence their use is halted for safety reasons. Nevertheless, they present interesting properties for their use as non-propagating vectors to deliver therapeutic genes in the TME, since they induce high expression levels of the therapeutic gene, apoptosis of the infected cells, and a strong local inflammation, mimicking the mechanism of action of an OV but in a more limited manner.

In general terms, immune-excluded or immune-desert tumors (also called “cold” tumors) correlate with lower response rates to ICB [106] and may benefit greatly from the simultaneous activation of more than one compartment of the immune system. In this sense, the use of OVs or self-amplifying RNA vectors could be an appealing option to increase immune cell infiltration and release of TAAs while unleashing the immune system through locally expressed ICIs (Figure 2). On the other hand, immune-inflamed tumors (or “hot” tumors) are more often associated with better response to ICB [106], hence the use of non-propagating vectors that provide an in situ expression of ICIs could be potentially effective at reinvigorating endogenous antitumor immune responses (Figure 2).

### 5.1. Oncolytic Viruses

Cancer therapy with oncolytic viruses (OVs) became a reality with the U.S. Food and Drug Administration (FDA) approval of T-VEC (talimogene laherparepvec) for the treatment of unresectable metastatic stage IIIB/C–IVM1a melanoma in 2015 [107]. T-VEC is based on an attenuated herpes simplex virus (HSV) that has been genetically modified to express granulocyte-macrophage colony-stimulating factor (GM-CSF). Treatment of melanoma patients with this virus has resulted in a 31.5% response rate with 16.9% complete responses. The antitumor effect of T-VEC is especially strong in accessible, treated lesions but becomes weaker in non-treated lesions and metastases. Interestingly, several clinical trials have shown that the combination of local tumor treatment with T-VEC and systemic administration of ICIs, such as anti-CTLA-4 and anti-PD-1 mAbs, results in increased antitumor efficacies [108,109]. Perhaps inspired by these last studies, many groups have focused on trying to simplify this combination therapy by engineering HSV vectors to express ICIs. Some of these “optimized” HSV OVs have been tested in preclinical models of GI tumors, such as CRC and liver cancer [110,111,112]. A rationale for expressing ICIs from OVs locally is that they are not completely able to overcome the immunosuppressive TME, in part because the viral infection can upregulate PD-L1 in tumors. For that reason, several HSV vectors have been engineered to express anti-PD-1 antibodies, either as scFvs [110,112] or nanobodies [111]. In fact, an HSV virus expressing an anti-PD-1 scFv was very efficient at modifying the TME in a liver cancer model based on Hepa 1–6 cells, improving antigen presentation by dendritic cells (DCs), and leading to better antitumor efficacy compared with the control virus [110]. The efficacy of this vector was improved when combined with systemic administration of anti-TIGIT [110], anti-CTLA-4, and anti-TIM-3 mAbs [112]. The large size of the HSV genome and the deletion of some viral genes, used to render the virus attenuated in humans, allow the generation of modified viruses harboring several transgenes. This offers the opportunity to express a combination of immunomodulators from the same vector, which in theory could potentiate antitumor responses. ONCR-177 is a modified oncolytic HSV armed with five different transgenes, including interleukin-12 (IL-12), Fms-related tyrosine kinase 3 ligand (FLT3LG), chemokine (C-C motif) ligand 4 (CCL4), a nanobody against PD-1, and a mAb against CTLA-4 [111]. This “super” oncolytic HSV has shown to be efficacious across a panel of syngeneic bilateral mouse tumor models, including some CRC tumors such as MC38 and CT26. These encouraging results have led to the clinical evaluation of ONCR-177 in patients with metastatic cancer (ONCR-177-101, NCT04348916).

A second type of OV that has been extensively evaluated for cancer treatment in both preclinical and clinical settings is based on modified adenoviruses. Although the oncolytic adenovirus (OAd) Oncorine was approved for the treatment of nasopharyngeal carcinoma in China in 2005, similar vectors such as Onyx-015 have not been authorized by the FDA or the European Medicines Agency (EMA). The main reason behind these decisions is that a limited therapeutic effect was observed in patients receiving Onyx-015. In an attempt to increase the potency of OAds, several groups have engineered these vectors to express ICIs [113,114,115]. Since a significant fraction of the population is seropositive for the most commonly used adenovirus vectors, such as human adenovirus serotype 5, an ingenious strategy is to generate vectors based on less prevalent serotypes, as it was conducted to produce some of the COVID-19 vaccines based on adenoviruses [116]. Following this approach, an oncolytic virus based on the chimpanzee adenovirus CV 68 was engineered to express a full-length anti-human PD-1 mAb [113]. This vector was quite efficient against MC38 tumors in a human PD-1 (hPD-1) knock-in mouse tumor model, having a similar effect to the combination of control virus and systemic anti-PD-1 mAb administration. As in the case of HSV, some groups have attempted to increase the potency of OAds by using several immunomodulatory genes. Given the reduced cloning capacity of OAds, an interesting approach for achieving this goal is to combine the propagating virus with a helper-dependent adenovirus (HD-Ad) vector, which is devoid of viral coding sequences and allows the inclusion of multiple transgenes [117]. Coadministration of OAd and HD-Ad allows the last one to be amplified and packaged in transduced cancer cells. This is the case of CAdTrio, which combines an OAd with a HD-Ad expressing a cytokine (IL-2), a checkpoint blocker (anti-PD-L1 mAb), and a bispecific tumor-targeted T-cell engager (BiTE) molecule recognizing CD3 on T cells and CD44 variant 6, a molecule widely expressed on tumors but not in normal tissues [115]. Although CAdTrio was not very efficient by itself, it showed potent antitumor effects when combined with HER2-specific chimeric antigen receptor (CAR) T cells in pancreatic and head and neck squamous cell carcinoma models. A final example of an ICI-armed adenovirus includes an oncolytic vector expressing a chimeric PD-L1 ligand based on a soluble PD-1 domain fused to the human Fc domains of immunoglobulin A1 (IgA1) and G1 (IgG1) [114]. This new molecule combined the ability to block PD-1/PD-L1 interaction with Fc-effector mechanisms mediated by IgA1 (neutrophil activation) and IgG1 (natural killer and complement activation). The OAd expressing this new ICI provided an enhanced antitumor effect in the CT26 colon cancer model compared with unarmed OAd or systemic administration of an anti-PD-L1 mAb with no effector functions (containing a mouse IgG2a isotype). In this case, engagement of natural killer (NK) cells by the human IgG1 Fc seemed to be important in tumor cell killing since depletion of CD8^+^ T cells did not abrogate the antitumor effect of this OAd in the 4T1 breast tumor model.

Another type of OV harboring a DNA genome that has been explored as an agent for cancer therapy is based on poxviruses. The high safety record of the vaccinia virus (cowpox), which has been used for many decades for smallpox vaccination leading to the global eradication of this disease [118], has prompted its evaluation in cancer therapy. In the context of GI tumors, two oncolytic poxviruses armed with ICIs have been tested. The first one was designed to co-express an anti-PD-L1 mAb and the human sodium iodide symporter (hNIS) and was able to increase overall survival in a model of human pancreatic cancer in nude mice when administered systemically [119]. A second vaccinia virus was armed with an anti-TIGIT mAb and was able to reshape the immunosuppressive TME from “cold” to “hot” status in several models of GI tumors, including subcutaneous CT26 and MC38 colon cancer nodules as well as an HCC ascites tumor model based on H22 cells [120].

Though not many oncolytic RNA viruses armed with ICIs have been tested in preclinical tumor models, a special mention must be made of the measles virus. This OV was one of the first engineered viruses to express mAbs against immune checkpoints (CTLA-4 and PD-L1), initially tested in a melanoma model [121]. More recently, the measles virus expressing an scFv against PD-L1 showed a remarkable effect in MC38 tumors, although it was lower than the one achieved with a similar vector expressing IL-12 [122]. Despite not having been tested, it is possible that the combination of IL-12 and anti-PD-L1 scFv expressed from the same measles vector could result in an even more potent effect.

### 5.2. Self-Amplifying RNA Vectors

Vectors with self-amplifying capacity are attractive tools for local cancer immunotherapy as they can provide high and transient expression of therapeutic molecules while inducing the death of the infected cell and stimulating innate immunity. These vectors are based on alphaviruses, such as Sindbis virus, Semliki Forest virus (SFV), and Venezuelan equine encephalitis virus [123]. Alphaviruses are enveloped arboviruses that belong to the *Togaviridae* family and are usually transmitted by mosquito vectors between vertebrate hosts, in which they induce diseases of different severity [124]. In general, alphaviruses have a broad tropism and can infect many different cell types, although the mechanisms of viral entry are not fully understood [125].

The genome of alphaviruses is a positive-sense single-strand RNA, approximately 11–12 kb in length. It mimics cellular messenger RNA (mRNA) since it contains a 5′ methylguanylate cap structure and a 3′ polyadenylate sequence. There are two open reading frames (ORFs) in the genome: Rep, of around 7 kb and next to the 5′ terminus, encodes for four non-structural proteins that constitute the viral replication complex, and a second ORF that encodes for the structural proteins [124]. Translation of Rep from the genome occurs directly in the infected cell, generating a large polyprotein (unprocessed replicase) that can synthesize the complementary negative-sense viral genomic RNA during the first stage of infection. Later, Rep is processed, and its mature form amplifies the viral genome using the minus-sense genome as a template, and it also amplifies a smaller plus-sense RNA from a sub-genomic promoter located upstream of the second ORF. The subgenomic RNA also mimics an mRNA, and it is translated to produce a large structural polyprotein that will give rise to the viral capsid and envelope proteins [126].

The main bottleneck for the clinical translation of alphaviruses is their broad tropism, which can have deleterious effects on healthy tissues. To increase their biosafety, the structural proteins are usually replaced by the therapeutic gene, generating vectors that are not able to propagate in vivo [127,128]. In addition, local vector administration in the tumor mass potentially limits the infection to cancer cells or tumor-associated cells. With this system, a high and transient expression of the therapeutic gene can be achieved thanks to the self-amplifying nature of the RNA, which promotes the death of the infected cell after 24–72 h. In addition, these vectors induce an important mobilization of the immune system, generating strong IFN-I responses due to the amplification of the viral RNA [123,129]. Therefore, these vectors could be useful to overcome the strong immunosuppression in the TME as well as to promote the release of TAAs, synergizing with other immunotherapies (Figure 2). The potential and safety of this system have been recently shown in a clinical trial using Vvax001, a therapeutic cancer vaccine consisting of a replication-incompetent SFV vector encoding human papillomavirus (HPV)-derived antigens E6 and E7 [130].

SFV vectors encoding immunostimulatory cytokines, such as IL-12, have shown strong antitumor activity in different preclinical models of GI tumors [131,132,133]. The antitumor potential of this vector in the context of ICB has also been shown by our group. In the work by Ballesteros-Briones et al., we demonstrated the potent antitumor activity of an SFV vector encoding an anti-PD-L1 antibody (SFV-aPDL1) in mouse models of CRC and melanoma [134]. Interestingly, the short-term expression of the anti-PD-L1 antibody (which was undetectable at day five post-SFV-aPDL1 administration) was sufficient to induce potent and long-lasting antitumor responses, promoting tumor-specific CD8^+^ T-cell infiltration in both tumor models. Upregulation of interferon-stimulated genes was also observed in CRC tumors treated with SFV-aPDL1 and a control vector encoding for the β-galactosidase gene (SFV-LacZ) [134]. IFN-I responses have been shown to be crucial for the antitumor effect of the SFV vector [135].

More recently, we have constructed SFV vectors encoding nanobodies against PD-1 and PD-L1 [136]. Potential advantages of using smaller antibody fragments include improved intratumoral distribution, higher levels of expression in vivo, and the possibility of generating different nanobody-based chimeric proteins. In this study, we observed modest therapeutic benefit when monomeric nanobodies were expressed from SFV vectors, presumably due to leakage of these small molecules outside the TME and rapid elimination from the bloodstream through renal clearance [137]. Taking advantage of the simplicity of genetic manipulation of single-domain antibodies, we fused them to the Fc domain of mouse IgG, generating larger, homodimeric molecules with improved PD-1/PD-L1 inhibition capacity. This strategy has already been used to optimize nanobodies in several preclinical studies and in some candidates that have reached clinical trials [138]. SFV vectors encoding these nanobody-Fc fusion proteins showed potent antitumor activity in mouse models of CRC and melanoma, outperforming control vectors encoding conventional antibodies against PD-1 and PD-L1 [136]. The superior performance of SFV vectors encoding nanobody-Fc constructs could be explained, at least in part, by the significantly higher levels of expression that were achieved compared to conventional antibodies, suggesting that lower doses of the vector could be used in clinical trials.

### 5.3. Non-Replicating Vectors

Most studies based on local expression of immunomodulatory mAbs have utilized OVs, given the additional antitumor activity of these vectors. However, despite the attractive properties of OVs for cancer therapy, these agents present some drawbacks, mainly related to their high immunogenicity. In fact, immune recognition of viral proteins, produced at high levels in infected tumor cells, leads to both humoral and cellular responses that limit the persistence of these viruses in vivo [139]. Therefore, when OVs are armed with therapeutic proteins, such as mAbs, their expression is usually short-term. On top of that, immunostimulatory molecules expressed by the virus can enhance immune responses against the vector itself, further limiting the duration of expression. In this sense, non-replicating viral vectors and non-viral vectors could provide a longer expression window in vivo due to their lower immunogenicity. Regarding non-replicating viral vectors, the ones that have been used more often in preclinical studies are based on adenovirus and adeno-associated virus (AAV).

One of the first attempts to deliver immunomodulatory mAbs locally into tumors was achieved with a first-generation adenovirus vector able to express an anti-CTLA-4 antibody [140]. Adenovirus vectors have several advantages, such as high transduction of tumor cells and the induction of innate and adaptive immune responses that contribute to the creation of proinflammatory TME and facilitate the activation of tumor-specific T cells. The adenovirus vector expressing anti-CTLA-4 mAb led to a significant delay in tumor growth, although this was only observed in combination with systemic Treg depletion. Despite the fact that the duration of anti-CTLA-4 mAb expression was not evaluated in this study, it is likely that it did not last long given that cells infected by first-generation adenoviruses are eliminated by the immune system due to the expression of viral proteins [141]. An interesting strategy to prolong the expression of adenovirus vectors is the use of HD-Ad, which have reduced immunogenicity as they are depleted of all viral genes. These high-capacity vectors allow the cloning of up to 35 kb of foreign DNA, making it possible to express large transgenes or complex regulatory systems [142]. This approach has been recently used by our group to deliver an anti-PD-L1 mAb to tumors based on the MC38 cell line [143]. The expression of anti-PD-L1 mAb from this vector was regulated by a mifepristone-inducible promoter, which allowed for tight control of mAb levels in vivo. This vector was able to induce potent antitumor responses in subcutaneous tumors but was less efficient in a more stringent model of progressing peritoneal carcinomatosis based on the same tumor cells. However, in the same vein as the observations from the adenovirus vector expressing anti-CTLA-4 mAb, depletion of macrophages led to an increase in therapeutic efficacy, highlighting the importance of tackling immunoregulatory mechanisms to overcome resistance to ICIs.

Another approach to obtain sustained in vivo expression of mAbs is the use of AAV vectors, since they are also deprived of all viral genes and do not lead to immune responses against infected cells. AAV vectors have proven to be one of the safest and most effective gene therapy vectors, with many candidates already tested in clinical trials and four products approved for human use [144]. Several studies performed by us and others have made use of AAV vectors to express immunomodulatory antibodies to treat tumors [145,146,147,148]. Two of these studies addressed the effect of AAV vectors systemically expressing an scFv [146] or a nanobody [145] against PD-1 in colorectal tumor models. In both cases, a single AAV administration was able to protect against the growth of MC38 tumors, although the vectors were administered either before or only two days after tumor cell inoculation. In both studies, maximum serum expression was reached several weeks after AAV administration, which probably limits the potency of these vectors against established tumors, at least in mouse models where tumor nodules grow rather quickly. Although in these studies the AAV vector was not administered intratumorally, other groups have evaluated the possibility of local vector administration in order to concentrate ICI expression in malignant tissues [147,148]. In particular, an AAV vector expressing an anti-PD-1 scFv fused to an Fc domain (AAV-αPD1) was tested in glioblastoma and renal carcinoma tumors expressing HER2. In these studies, specific delivery of the vector to tumor cells was achieved by displaying a HER2-specific ankyrin repeat protein (DARPin) on the AAV surface. Since HER2 is a tumor antigen frequently expressed in some GI tumors, such as pancreatic and gastric cancer, this strategy could also have a broader therapeutic use. Nevertheless, AAV-αPD1 only showed modest antitumor activity when combined with other therapeutic strategies, such as chemotherapy [147] or HER2-specific CAR NK cells [148].

### 5.4. Non-Viral Vectors

An attractive alternative to viral vectors is the use of nucleic acids, in the form of DNA or RNA, to deliver ICIs to tumors. Non-viral vectors can be easily designed and manufactured at GMP level and provide a high safety profile since no virus is present in these preparations. An additional advantage of non-viral vectors is that no humoral responses are elicited against the vector, allowing the use of repetitive doses, something that is usually not feasible for viral vectors due to the generation of neutralizing antibodies that limit the effect of subsequent doses. Furthermore, the absence of packaging restrictions in DNA and RNA vectors allows the delivery of large genes or expression cassettes, which is particularly interesting for mAb genetic constructs. Although the possibility to use DNA and RNA to deliver therapeutic genes in vivo has been explored for many years, it has not become clinically feasible until recently, due to the development of new devices for in vivo electroporation and improved lipid nanoparticle (LNP) formulations able to efficiently encapsulate and deliver nucleic acids, such as the ones currently used for COVID-19 mRNA vaccines [149].

Intratumor electroporation of plasmid DNA expressing ICIs has been evaluated in several preclinical models of CRC, showing antitumor activity [150,151,152]. In general, the best results were obtained when tumors were electroporated with a plasmid expressing an anti-CTLA-4 mAb [150,151], although the efficacy was increased by combining this plasmid with a second one expressing an anti-PD-1 mAb [150]. A further enhancement of therapeutic responses was obtained by co-electroporating tumors with a third plasmid encoding IL-12 [152]. The possibility of combining several plasmids in the same treatment makes this approach very versatile for cancer treatment.

The use of RNA to deliver ICIs to GI tumors has also been explored by the administration of LNPs containing an mRNA encoding for pembrolizumab (an anti-hPD-1 mAb) in MC38 tumors implanted in an hPD-1 knock-in mouse model [153]. This treatment, which was given intravenously, led to a delayed growth of intestinal tumors and an improvement in survival.

Finally, alphaviruses are also highly attractive tools to be used as non-viral vectors since their self-amplifying ability is retained even when they are delivered as nucleic acids. Similar to viral vectors, they are able to induce high levels of transgene expression and apoptosis in the transfected cell. Non-viral vectors based on alphaviruses could be used in the form of RNA directly or as DNA, in which the self-amplifying RNA is cloned under the control of a eukaryotic promoter allowing its transcription in the target cell [154,155]. We have previously demonstrated the remarkable antitumor potential of these strategies by delivering SFV RNA and DNA vectors expressing IL-12 or an anti-PD-1 nanobody, respectively, by electroporation in MC38 subcutaneous tumors [136,156]. In addition to the adjuvant effect generated by the immunogenic death of transfected cells, this system would theoretically allow the use of lower vector doses to reach equivalent levels of transgene expression compared to conventional mRNA or DNA vectors [154].

## 6. Conclusions and Future Perspectives

Since the discovery of immune checkpoints, many groups have addressed the possibility of using gene therapy vectors to express ICIs in vivo and, in particular, locally in tumors (a summary of the most relevant strategies covered in this review is shown in Table 1). The rationale behind this approach is that the use of a genetic vector could provide a continuous endogenous source of the blocking antibody without the need to give repeated injections to the patients. In addition, local expression provided by the vector could have a higher safety profile and limit systemic toxicities. As discussed in this review, this approach has shown to be successful in many different types of preclinical models of GI tumors, including CRC, HCC, and pancreatic tumors, using different types of vectors. The combination of local therapy using OVs and systemic administration of ICIs has already been tested in clinical trials, with excellent results obtained when using T-VEC and anti-PD-1 or anti-CTLA-4 antibodies. However, new trials will be needed to evaluate the efficacy and safety of local ICI expression in tumors. Given the higher safety profile of non-viral vectors, it is likely that these will be the first ones to be tested in accessible tumors such as melanoma, where local treatment based on electroporation of a plasmid encoding IL-12 has already shown very promising results [157]. However, although they could provide longer transgene expression, non-viral vectors lack the adjuvant effects provided by viral components, especially in the case of OVs, making them less potent to trigger antitumor responses quickly. Local tumor treatment with vectors expressing ICIs may need to be complemented with systemic treatment using the same or a different ICI. Although this last approach may seem redundant, it could allow to (i) increase abscopal effects from local treatment, (ii) reduce the amount of ICI to be given systemically, and (iii) attenuate the high toxicity observed in patients treated with two different ICIs, such as anti-PD-1 and anti-CTLA-4, since one of them will be expressed only locally. Finally, given the influence that microbiota composition can have on immunotherapies against GI tumors, it is possible that local delivery of ICIs by gene therapy vectors could also benefit from strategies aimed at modulating the gut microbiota, something that has not been explored so far.

## Figures and Tables

**Figure 1 cancers-15-02352-f001:**
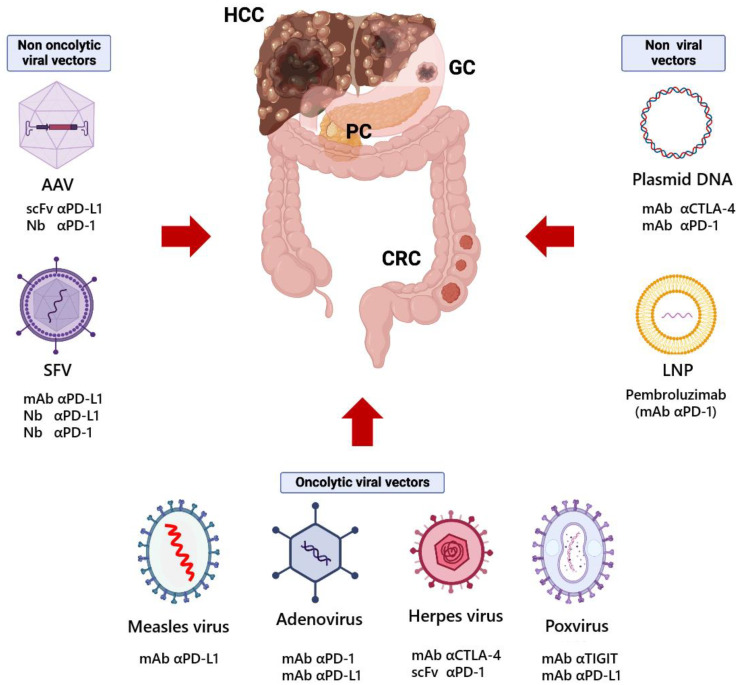
Gene therapy vectors used for the expression of antibodies against immune checkpoints in preclinical models of gastrointestinal cancers. HCC, hepatocellular carcinoma; GC, gastric cancer; PC, pancreatic cancer; CRC, colorectal cancer; AAV, adeno-associated virus; SFV, Semliki Forest virus; LNP, lipid nanoparticle; mAb, monoclonal antibody; Nb, nanobody; scFv, single-chain variable fragment; α, anti. This figure was created using BioRender.com.

**Figure 2 cancers-15-02352-f002:**
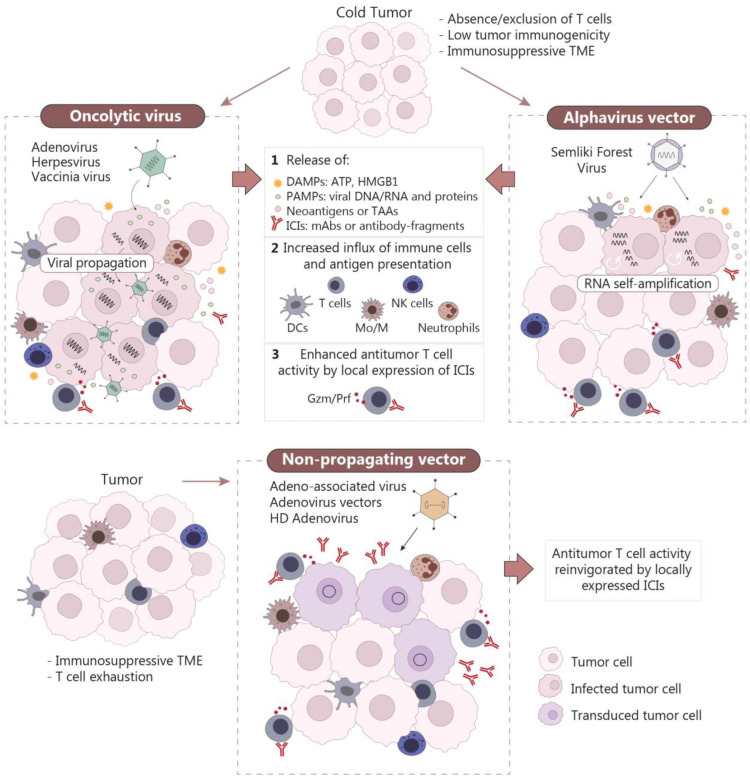
Mechanisms of action of gene therapy vectors encoding antibodies against immune checkpoints. Oncolytic viruses and alphavirus vectors based on self-amplifying RNA are able to induce inflammation in tumors through the release of PAMPs (pathogen-associated molecular patterns) and DAMPs (damage-associated molecular patterns) due to the immunogenic death of tumor cells. These processes favor the infiltration of immune cells in the tumor, which can be further activated by locally expressed ICIs (**upper panel**). Non-propagating vectors are usually less immunogenic; however, local delivery of ICIs by these vectors can promote the activation of tumor-infiltrating immune cells (**lower panel**). TME, tumor microenvironment; HMGB1, high mobility group box 1; TAAs, tumor-associated antigens; ICIs, immune checkpoint inhibitors; DCs, dendritic cells; Mo/M, monocytes/macrophages; NK, natural killer; Gzm/Prf, granzyme/perforin; HD, helper-dependent.

**Table 1 cancers-15-02352-t001:** Relevant preclinical studies based on ICI expression from gene therapy vectors.

Cancer Type	Vector	Gene(s)	Results	Reference
CRC	HSV	αCTLA-4 mAb, αPD-1 Nb, IL-12, CCL4, FLT3LG	Inhibition of tumor growth and improved survival	[111]
Liver cancer		αPD-1 scFv	Promotion of antitumor immunity and synergy with TIGIT blockade	[110]
			Systemic antitumor memory response and synergy with CTLA-4 and TIM-3 blockade	[112]
CRC, breast and lung cancer	OAd	sPD-1-IgGA Fc	Increased efficacy in tumor cell killing	[114]
CRC		αPD-1 mAb	Inhibition of tumor growth and improved survival	[113]
Pancreatic and head and neck cancer	OAd+ HD-Ad	αPD-L1 mini-antibody, αCD44v6 BiTE, IL-2	Synergistic effect with αHER2-CAR T cells	[115]
CRC	HD-Ad	αPD-L1 mAb	Controlled expression and potent antitumor activity	[143]
	Measles virus	αPD-L1 scFv	Inhibition of tumor growth and improved survival	[122]
Pancreatic cancer	Ortho- poxvirus	αPD-L1 mAb, hNIS	Inhibition of tumor growth and improved survival	[119]
CRC	Vaccinia virus	αTIGIT scFv	Promotion of antitumor immunity and synergy with PD-1 and LAG-3 blockade	[120]
CRC and melanoma	SFV	αPD-L1 mAb	Inhibition of tumor growth and improved survival	[134]
		αPD-1 Nb-Fc, αPD-L1 Nb-Fc	Superior antitumor activity than SFV vectors expressing mAbs	[136]
CRC	AAV	αPD-1 Nb	Protection against tumor challenge	[145]
CRC and breast cancer		αPD-L1 scFv	Relief in immunosuppression, tumor growth control and improved survival	[146]
Her2/neu tumors	Her2-AAV	αPD-1 scFv-Fc, Nivolumab	Local ICI expression by tumor-targeted AAV vector	[147]
CRC	DNA plasmids	αCTLA-4 mAb, αPD-1 mAb	Synergistic antitumor effect	[150]
	LNP-encapsulated mRNA	Pembrolizumab	Inhibition of tumor growth and improved survival	[153]
	SFV DNA plasmid	αPD-1 Nb-Fc	Similar antitumor effect than SFV viral particles expressing αPD-1 Nb-Fc	[136]

CRC, colorectal cancer; HSV, herpes simplex virus; OAd, oncolytic adenovirus; HD-Ad, helper-dependent adenovirus; SFV, Semliki Forest virus; AAV, adeno-associated virus; Her2-AAV, receptor-targeted AAV vector binding to the tumor antigen Her2/neu; LNPs, lipid nanoparticles; α, anti; mAb, monoclonal antibody; Nb, nanobody; scFv, single-chain variable fragment; sPD-1, soluble PD-1; BiTE, bispecific T-cell engager.

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
