# Peer review of "Local Delivery of Immunomodulatory Antibodies for Gastrointestinal Tumors"

_cancers, 2023, doi:10.3390/cancers15082352_

Round 1
Reviewer 1 Report
The Manuscript of Noelia Silva-Pilipich, Ángela Covo-Vergara and Cristian Smerdou is very interesting and well written and describes the delivery systems (gene therapy vectors like oncolytic viruses, self-replicating RNA vectors, and non-replicative viral and non-viral vectors) used for immunomodulatory antibodies targeting gastrointestinal tumors. These innovative therapeutic strategies mainly target one of the hallmarks of cancer: immune evasion. The authors should answer to the following minor requests to improve their Manuscript.
MINOR REVISIONS
1)In the line 192 of their Review, the authors indicated the dietary intervention as a possible strategy to increase the efficacy of the therapies for gastrointestinal tumors based on antibodies which target CTLA-4, PD-1, PD-L1 (among the other molecular markers). Which are the natural molecules that can be taken with the diet and that can increase the efficacy of immunomodulatory antibodies?
2)The authors should discuss the use of human red blood cells as efficient delivery system for antibodies used for gastrointestinal tumors (in particular for colorectal cancer), as already described for the FAT1-specific monoclonal antibody mAb198.3 [Grifantini, R. et al., 2018; DOI: 10.1016/j.jconrel.2018.04.052 ]. Could the human red blood cells be suitable for the delivery of antibodies targeting CTLA-4 and PD-1 in gastrointestinal cancers?
Author Response
MINOR REVISIONS
1) In the line 192 of their Review, the authors indicated the dietary intervention as a possible strategy to increase the efficacy of the therapies for gastrointestinal tumors based on antibodies which target CTLA-4, PD-1, PD-L1 (among the other molecular markers). Which are the natural molecules that can be taken with the diet and that can increase the efficacy of immunomodulatory antibodies?
We have now extensively revised section 3 of the review (Microbiota impacts on immunotherapy efficacy) and have included a paragraph with information about molecules that can be taken with the diet and other dietary interventions able to increase the efficacy of immunomodulatory antibodies (lines 487-499). One such compound is pectin, a widely consumed soluble fiber, which was able to restore anti-PD-1 efficacy in tumor-bearing mice humanized with a patient’s gut microbiota by beneficially regulating its composition and diversity.
2)The authors should discuss the use of human red blood cells as efficient delivery system for antibodies used for gastrointestinal tumors (in particular for colorectal cancer), as already described for the FAT1-specific monoclonal antibody mAb198.3 [Grifantini, R. et al., 2018; DOI: 10.1016/j.jconrel.2018.04.052 ]. Could the human red blood cells be suitable for the delivery of antibodies targeting CTLA-4 and PD-1 in gastrointestinal cancers?
We thank the reviewer for this suggestion and have now included a comment about this interesting strategy to deliver mAbs in section 4 (see lines 630-637 and new reference 94).
Reviewer 2 Report
This is an excellent, up-to-date review of gene therapy as local therapy to deliver immune checkpoint inhibitors based on antibodies to gastrointestinal cancers. The paper can be published as it is with no further modifications.
Author Response
We thank the reviewer for the positive feedback
Reviewer 3 Report
This review on immunomodulation therapy for gastrointestinal cancers is interesting, and this reviewer read it with avid interest. While similar reviews have been published over the year, regularly, this specific manuscript is fine, well-written, and well-documented.
The manuscript by itself is fine. Only minor English language editing and style should be required, especially for the abstract.
Main comments: The immunomodulation and immunotherapy for gastrointestinal tumors have been the subject of several reviews over the last few years. In the abstract and introduction, the authors should make clearer the added value of this review with the latest basic and clinical research data on the subject.
Abstract: Compared to the main text, the abstract fell out of place or felt like it was written hastily. It would be best to revise the abstract for better conciseness and clarity.
Minor comments Main text: Minor editing is required for typos and grammar inconsistencies. Figures 1 and 2: Minor details, but please increase the font size.
Author Response
Main comments:
The immunomodulation and immunotherapy for gastrointestinal tumors have been the subject of several reviews over the last few years. In the abstract and introduction, the authors should make clearer the added value of this review with the latest basic and clinical research data on the subject.
We have now made clearer in both the abstract and introduction the added value of this review. Specifically, we think that the main novelty of this review is that it covers the use of gene therapy vectors to deliver checkpoint inhibitors for gastrointestinal tumors. To our knowledge there are no recent reviews on this subject. The following sentence has been added in the abstract: “This review covers the most recent studies related to the use of gene therapy vectors to deliver ICIs to gastrointestinal tumors”. In this line we also included in the introduction a sentence highlighting the added value of this review: “In this review, we describe the latest developments in the field of gene therapy to deliver ICIs based on antibodies to GI tumors and comment on combination strategies that are being evaluated with ICB therapies”.
Abstract: Compared to the main text, the abstract fell out of place or felt like it was written hastily. It would be best to revise the abstract for better conciseness and clarity.
We have now revised the abstract and hope that it is clearer and more concise.
Minor comments
Main text: Minor editing is required for typos and grammar inconsistencies.
The manuscript has been revised and a few typos and grammar inconsistencies have been corrected.
Figures 1 and 2: Minor details, but please increase the font size.
We have now increased the font size in both figures 1 and 2.